# Association between Hepatocellular Carcinoma Recurrence and Graft Size in Living Donor Liver Transplantation: A Systematic Review

**DOI:** 10.3390/ijms24076224

**Published:** 2023-03-25

**Authors:** Alessandro Parente, Hwui-Dong Cho, Ki-Hun Kim, Andrea Schlegel

**Affiliations:** 1HPB and Transplant Unit, Department of Surgical Science, University of Rome Tor Vergata, 00133 Rome, Italy; 2Division of Hepatobiliary and Liver Transplantation, Department of Surgery, Asan Medical Center, University of Ulsan College of Medicine, Seoul 05505, Republic of Korea; 3Fondazione IRCCS Ca’ Granda, Ospedale Maggiore Policlinico, Centre of Preclinical Research, 20122 Milan, Italy; 4Department of Surgery and Transplantation, Swiss HPB Centre, University Hospital Zurich, 8091 Zurich, Switzerland

**Keywords:** living donor liver transplantation, hepatocellular carcinoma, graft-to-recipient weight ratio, graft size, tumor recurrence

## Abstract

The aim of this work was to assess the association between graft-to-recipient weight ratio (GRWR) in adult-to-adult living donor liver transplantation (LDLT) and hepatocellular carcinoma (HCC) recurrence. A search of the MEDLINE and EMBASE databases was performed until December 2022 for studies comparing different GRWRs in the prognosis of HCC recipients in LDLT. Data were pooled to evaluate 1- and 3-year survival rates. We identified three studies, including a total of 782 patients (168 GRWR < 0.8 vs. 614 GRWR ≥ 0.8%). The pooled overall survival was 85% and 77% at one year and 90% and 83% at three years for GRWR < 0.8 and GRWR ≥ 0.8, respectively. The largest series found that, in patients within Milan criteria, the GRWR was not associated with lower oncological outcomes. However, patients with HCC outside the Milan criteria with a GRWR < 0.8% had lower survival and higher tumor recurrence rates. The GRWR < 0.8% appears to be associated with lower survival rates in HCC recipients, particularly for candidates with tumors outside established HCC criteria. Although the data are scarce, the results of this study suggest that considering the individual GRWR not only as risk factor for small-for-size-syndrome but also as contributor to HCC recurrence in patients undergoing LDLT would be beneficial. Novel perfusion technologies and pharmacological interventions may contribute to improving outcomes.

## 1. Introduction

Hepatocellular carcinoma (HCC) is the most common primary liver cancer with increasing incidence and estimated death rate of 55% until 2040 [1,2]. Since the development of the Milan criteria in 1996, liver transplantation (LT) has been accepted as main curative liver cancer treatment [3]. In countries with no or limited access to a deceased donor organ pool for various reasons, living donor liver transplantation (LDLT) has been developed with excellent outcomes for HCC recipients.

Some early reports demonstrated advantages for LDLT over deceased donation, when considering patients with HCC, findings that were further supported by several prospective intention-to-treat analyses, where the overall recipient survival was comparable between living and deceased liver recipients [4,5]. A few other studies made such findings with even better recipient survival rates after LDLT, indicating that liver regeneration processes might be of limited relevance for HCC recurrence [6,7].

Another inherent advantage appears with the shorter waiting times for candidates where a living donor is available. Goldaracena et al. demonstrated that patients who had a potential live donor at the time of listing had a higher survival rate [6]. Interestingly, the authors found that waiting times of 9–12 months or ≥12 months were predictors of death [6]. In another study, Lai et al. showed that that having a potential live donor graft could decrease the intention-to-treat risk of death in patients with HCC who are on a waiting list for a liver transplant, due to fewer dropouts from the waiting list even in centers where both live donation and decease donation options are equally available [8].

Despite such results and the general opinion that LDLT imposes a lower risk and better outcomes in liver recipients, some other authors have also reported controversial data. Fisher et al. presented inferior oncological outcomes after LDLT compared to deceased donor transplantation, as patients undergoing LDLT had higher HCC recurrence rates within 3 years [9]. Such findings were later supported by another analysis, where authors observed a higher recurrence after LDLT, which could have been due to different tumor characteristics and HCC management before transplantation [10]. In 2013, Gant et al. provided a systematic literature review and meta-analysis and reported an inferior disease-free survival (DFS) after LDLT when compared to deceased donation [11]. The literature underlines the need for and improved study design and reporting in future trials to understand if the observed DFS difference might be attributed to a study bias or are the result of other contributors, as seen in context of LDLT [11].

The overall controversial clinical results also point to confounders, other than recipient cancer risk, that may have impact on recurrence rates. Donor risk factors (i.e., prolonged warm or cold ischemia), surgical parameters and stress (duration of transplantation, medical management) and the recipient risk (medical fitness, lab MELD score) are just a few adjunct variables, which add to the already evident recurrence risk conveyed by the candidate’s liver cancer status and biology [12,13]. Such parameters may directly impact on the liver tissue quality and subsequently on the ability of the new liver to recover and regenerate after transplantation [14].

Despite the generally better quality of living donor liver grafts, compared to the deceased donor pool, the overall strategy is to protect the donors and to avoid the development of small-for-size-syndrome (SFSS). Per definition the development of a SFSS represents a situation where a small graft shows a primary dysfunction within the first postoperative week after transplantation without any sign of other pathologies (i.e., vascular complications, bile leak, sepsis) [15]. While the majority believe in the distinct clinical entity of a SFSS, the ongoing debate on underlying mechanisms involves portal hyper-perfusion along with a form of outflow obstruction [16]. This terminology appears to be somewhat misleading because the graft does not necessarily need to be small; SFSS-features can also occur when the liver tissue quality is impaired (i.e., steatosis) or when a partial liver is exposed to portal hypertension, as often seen in candidates with advanced liver disease staged as Child–Pugh grade C [16].

Additional risk factors may contribute to the observed SFSS, including the transplanted liver graft volume and the cytokine release triggered by both liver transection during donation surgery and after reperfusion [17]. The mechanistic link between an advanced hepatic ischemia-reperfusion injury (IRI) and liver tumor regrowth and metastasis was previously demonstrated in 2007 by Man et al., who described higher HCC recurrence rates and more lung metastases when a small liver remnant was evident [17,18].

Based on the above-mentioned concerns, the aim of this study was to perform a systematic literature review of all studies that reported data on oncological outcomes after LDLT for HCC and provide details on the association between liver graft rate recipient weight ratio (GRWR) and oncological outcomes.

## 2. Results

### 2.1. Baseline Characteristics and Demographics

A total of 45 papers were identified and screened. In 16 studies, the vast majority of authors reported their survival and SFSS-rates after LDLT using a cut-off for GRWR at <0.8% vs. ≥0.8% (Table 1) [19,20,21,22,23,24,25,26,27,28,29,30,31,32,33,34]. Most studies report a lower DFS when the liver volume is lower (i.e., GRWR < 0.8) compared to higher volumes. Centers from specific countries in Asia, including Japan, Hong Kong and India (Table 1), report the routine transplantation of living donor grafts with a smaller GRWR of <0.7 or even <0.6 in a very selective recipient cohort avoiding any additional risk factor, due to the higher risk of developing a SFSS. Despite the good overall pool of available data on the topic explored here, the number of studies transparently demonstrating the HCC risk factors together with liver GRWRs and acceptable follow-up duration after LDLT is very few.

The included studies reported a total of 782 patients; 168 underwent LDLT with a GRWR of <0.8%. Conversely, 614 were recipients, who received a liver with a GRWR of ≥0.8%. All patients were diagnosed with an HCC.

The mean donor age ranged between 26 and 33 years. The recipient age was found to be between 47 and 54 years, with the majority being male (range 76–98%), in the overall cohort. The baseline characteristics are summarized in Table 2. 

From this pool of literature, three studies met the inclusion criteria (Figure 1). Such studies were all retrospective and published between 2007 and 2018 [35,36,37].

### 2.2. Graft Characteristics

Three studies reported the graft types [35,36,37]. In the GRWR ≥ 0.8% cohort, there were 554 right lobes, 19 left lobes and 30 dual grafts. Conversely, there were 141 right lobes, 23 left lobes and 3 dual lobes in the GRWR < 0.8 group. The mean graft weight was reported only by one study, and it resulted in 680 (615–743) gram and 555 (500–607.5) gram liver graft weight for the two groups, GRWR ≥ 0.8% and GRWR < 0.8%, respectively [36]. Most studies [35,36] did not specify which liver lobe was used for LDLT (left: LLG (H1234) or right: RLG (H5678) [38]. The strategy for the management of the middle hepatic vein is not discussed; therefore, we have added the new terminology using LLG (H1234) or right: RLG (H5678) [38].

Lee et al. [37] exclusively transplanted right liver lobes (RLG: H5678) and reported that any middle hepatic vein branches of >5 mm in diameter were saved and underwent reconstruction. If the inferior right hepatic vein was >5 mm in size measured after right hepatic vein anastomosis, the inferior right hepatic vein was also anastomosed to the inferior vena cava.

### 2.3. Tumor Characteristics

One study [36] reported the overall number of locoregional treatments before transplantation, reporting a higher rate (n = 108/239, 45%) for GRWR ≥ 0.8% in contrast to GRWR < 0.8% (n = 16/56, 28%). Vascular invasion was present in the majority of patients in both groups, as reported by two studies and ranged between 42–78%. The mean number of tumor nodules was one in two studies (Table 3). Preoperative alpha-feto protein (AFP) was reported by the same two studies and ranged between 13 and 231 ng/mL in the entire cohort. The accepted tumor criteria for liver transplantation were also different. Two studies [35,37] used Milan criteria and reported a number of 256/375 (68.2%) and 75/112 (66.9%) recipients inside Milan criteria for GRWR ≥ 0.8% and GRWR < 0.8%, respectively. The third study [36] focused on the Hangzhou criteria and the authors reported 134 (65.6%) and 17 (50%) patients within criteria for GRWR ≥ 0.8% and GRWR < 0.8%, respectively. The graft and tumor characteristics are summarized in Table 3.

### 2.4. Risk of Bias Assessment

The risk of bias assessment of the included studies is demonstrated in Table 4. Overall, the two studies were deemed to be of good quality given the scores from NOS system. On the other hand, one study was deemed to be fair quality based on a score of six from the NOS system.

### 2.5. Outcomes Analysis

#### 2.5.1. Overall Survival

One study [37] reported the 1-, 3- and 5-year overall survival (OS) rates, which were 87.8%, 80.3% and 78.7%, respectively, for patients with GRWR < 0.8%, and 93.5%, 87.1% and 84.1%, respectively, for patients with GRWR ≥ 0.8%. The other survival rates were extrapolated and are merged in Figure 2.

#### 2.5.2. Disease-Free Survival

One study reported [37] the 1-, 3- and 5-year disease-free survival rates which were 75.9%, 73.3% and 71.7%, respectively, for patients with GRWR < 0.8%, and 86.4%, 80.8% and 77.9%, respectively, for patients with GRWR ≥ 0.8%. The other survival rates were extrapolated and are merged in Figure 2.

## 3. Discussion

This systematic review focuses on patients undergoing living donor liver transplantation (LDLT) for hepatocellular carcinoma (HCC). The overall number of suitable studies that transparently report the graft volume together with the candidate’s HCC cancer status with enough follow-up is limited to three. LDLT using small grafts with a graft recipient weight ratio (GRWR) of <0.8% was associated with lower overall and particularly tumor-free survival rates in patients with higher tumor burden outside Milan criteria. Although there were only three studies with a relatively small number of cases, this systematic review points to an outcome benefit considering donor liver volume in HCC recipients, particularly in candidates outside standard tumor criteria. 

LDLT has emerged as standard treatment, especially in most Asian countries where the deceased donor transplant program is limited or not active [39]. LDLT offers the advantages of an elective procedure which can be planned in advance, and a promptly available graft, thus decreasing waitlist times, a key factor for better survival, particularly in candidates with liver cancers or metastases. However, some authors have shown that LDLT may increase recurrence rates with impaired disease-free survivals compared DDLT [40]. In fact, reducing waitlist times does not allow the prolonged evaluation of the tumor biology and thus prevent the selection of those patients, who would drop out in the waiting list. Patients that receive a selective LDLT may therefore have a more aggressive tumor biology, which could be the reason why in some studies LDLT was found to induce impaired survival and early HCC recurrence rates. In cohort studies, where LDLT and deceased donor transplants are well-matched for recipient tumor features and overall risk, comparable results are described in centers with experience in both techniques [41]. Additionally, shorter waiting times in context of LDLT are often beneficial in preventing the risk of rapid tumor progression with better overall and DFS rates.

Available intention-to-treat analyses have demonstrated a survival advantage for living donor liver recipients over DDLT [5,8,42,43]. Our study confirmed that most of the candidates benefit from LDLT, provided their HCC criteria are within standard profiles. However, a thorough graft selection for patients with an advanced tumor burden should be considered, particularly in cases where the calculated GRWR is expected to be <0.8%, because this sub-cohort is exposed to an elevated risk for impaired overall and tumor-free recipient survival rates as shown in the largest report [37]. Survival rates were substantially different comparing the here included studies. One possible explanation could be that most living donor grafts in the study by Hwang et al. [35] that had a GRWR of <0.8% were left lobes (H1234). In contrast, the majority of livers in the other studies were right lobes (H5678) [36,37]. Available studies assessing the impact of right or left living donor lobes on the development of SFSS or HCC recurrence are scarce. Additionally, it should be highlighted that the study of Hwang et al. was conducted between 1992 and 2004; therefore, the authors might have been careful in selecting donors and recipients; the medical management was probably different and surgical techniques have advanced since.

Although a GRWR of >0.8% is preferable, levels GRWR between of ≥0.6% and 0.8% can be acceptable if an alternative living donor is lacking, for primary transplants of recipients with maximal one organ failure. For this approach, both the donor and recipient should consent to accept the higher risk [34] and an independent, multi-disciplinary team for LDLT should be involved. The calculation of the GRWR is generally based on liver graft weight and recipient weight, which can be challenging due to the known recipient edema, ascites and pleural fluid in contexts of advanced liver cirrhosis. Such clinical features may lead to a false high body weight, requiring a false high liver volume, protecting the recipient and potentially exposing the donor to a higher risk [44]. Despite the findings in this systematic review that suggest the importance of being cautious with low liver volumes to avoid higher HCC recurrence rates, donor safety remains the highest priority. Various factors have been proposed to contribute to the development of SFSS, including the venous congestion, portal hypertension along with arterial spasms and hypoperfusion. In addition, an insufficient liver mass and metabolically impaired tissue quality contribute further [16]. Small grafts (GRWR < 0.8) demonstrated significantly higher portal venous pressures (PVP) in different case series [45,46]. In such series, SFSS-grafts demonstrated elevated PVP for up to 2 weeks after transplantation, which was in contrast to liver grafts with appropriate volumes. The PVP of >20 mmHg was found to be associated with an impaired graft survival after LDLT. Also of interest are experimental studies with the transplantation of small partial liver grafts [47,48,49]. An elevated PVP in the early posttransplant phase correlated with rapid liver hypertrophy along with low portal blood levels of VEGF and elevated peripheral HGF values. These features were predictive for graft dysfunction (with coagulopathy, ascites, hyperbilirubinemia) and an overall poor outcome (Figure 3). These studies did, however, also demonstrate that an adequate elevation of portal venous pressure and flow is beneficial to trigger liver regeneration.

The relevance of blood flow and shear stress is well-described as trigger of endothelial cell dysfunction in different vessels. Disturbed blood flows with subsequent reduced shear stress at the endothelial cells, as seen with relevant portal hypertension in small liver grafts, may significantly impact mitochondrial function and increase mitochondrial ROS release with inflammation, thereby blocking mitochondria from their essential task to produce ATP—the key molecule needed for the full metabolic function of a liver, which can quickly regenerate without severe SFSS [50].

Based on the increasing surgical experience during the past few decades, various surgical techniques to modulate the liver inflow now exist, including splenic artery ligation, splenectomy or portocaval shunts. Such surgical steps are often required when the GRWR is below 0.7%. Intraoperative decision making is based on the portal vein and hepatic artery flow and pressure measurements (Figure 3). Other factors including the level of portal hypertension play an additional role intraoperatively, but also during the planning of the initial strategy in context for the estimated GRWR when donor and recipient are matched before transplantation [51]. In addition to the best possible inflow modulation, the optimal liver outflow reconstruction includes the creation of large anastomoses to the cava vein directly or through interposition grafts. Most surgeons consider the diameter of >5 mm for hepatic vein branches as relevant to perform an additional reconstruction.

The Cleveland group has recently demonstrated that an augmented graft outflow reconstruction (i.e., all three recipient hepatic veins used for the outflow of LLG-LDLT) together with meticulous PVP/flow measurement and inflow modulation whenever appropriate (before or after graft implantation) can lead to excellent and comparable results after LDLT using small grafts (GRWR < 0.7) [51].

Although living donor grafts are carefully selected and of optimal tissue quality, cells are subjected to a few more minutes warm ischemia at the time of recovery. Considering that such grafts undergo several blood-inflow clamping cycles (Pringle maneuver) during the transection of the donor liver, which activates the IRI-cascade, there is always a level of IRI seen in the recipient after transplantation. In fact, any intervention with subsequent tissue trauma, including surgery may trigger hypoxia and inflammation. The period of hypoxia (i.e., during pringle in an entire lobe or more regional during liver positioning for a specific preparation during partial liver resection) leads to an anaerobic metabolism in mitochondria with dys- or nonfunction of the respiratory chain and subsequent accumulation of succinate and other metabolites [52]. At reperfusion with the reintroduction of oxygen (i.e., after liver graft implantation, or removal of the pringle or repositioning of the lobe) in mitochondrial complex I, reactive oxygen species (ROS) are released, triggered by the level of accumulated succinate.

These ROS release initiates further downstream sterile inflammation and with damage of the mitochondrial and cellular wall thereby releasing further molecules including danger-associated-molecular patters (DAMPS, i.e., Hmgb1, mtDNA, ATP) and various pro-inflammatory cytokines (i.e., TNF-α, IL-1b, IL-18) [12]. An ongoing sterile tissue inflammation also involves the macrophages and endothelial cells, which trigger another wave of inflammation when in contact with patient or recipient blood components such as neutrophils. Further ROS, DAMPS and cytokines are released and the entire cascade of IRI inflammation is maintained. An increasing body of literature exists demonstrating the link between mitochondrial injury and function, the release of ROS and mitochondrial DNA and features of ongoing inflammation in the liver periphery. Particularly, this microenvironment was found to promote the resettling, migration and growth of circulating cancer cells from candidates with typical liver tumors, including HCC and cholangiocarcinoma. A favorable tumor biology with less active and less aggressive HCC-types are therefore beneficial and results in better survivals together with the best possible liver preservation and surgical techniques that are performed by experienced centers triggering the least possible inflammation [12].

The ROS–DAMPS–cytokine cascade instigates further downstream effects in the liver microenvironment, enabling the migration of circulating tumor cells through the sinusoidal endothelial cells (SEC) barrier. In fact, during inflammation, SECs can swell and lose their integrity, thus increasing their permeability. Consequently, tumor cells can migrate, invade and grow. In addition to these mechanisms, hypoxia triggers the release of the hypoxia-inducible-factor, which promotes tumor cell proliferation, migration and angiogenesis, with secretion of vascular endothelial growth factor. Such molecular pathways are of high relevance in liver transplantation, in particular when additional donor and recipient risk factors are involved, such as marginal donors or smaller grafts.

Next to the maintenance of liver function and recovery from IRI-injury, mitochondria are essential components to produce enough energy for liver regeneration. Impaired mitochondria trigger delays in liver regeneration particularly described in patients suffering from high inflammation after liver resection or partial transplantation. Delayed hepatocyte cell cycle passages and delayed entrance into the cell cycle were described.

The ongoing mitochondrial ROS release and chronic inflammation can also damage healthy cells and their mitochondrial DNA, which may lead to new DNA mutations, and upregulate proto-oncogenes together with a downregulation of tumor-suppressor genes, and hence, the development of new cancers [12].

Centre experience, logistics, medical recipient fitness and the duration of cold ischemia are additional factors with effect on the IRI-severity, the subsequent liver function, regeneration, and recipient recovery. Triggered by an elevated IRI-associated inflammation and related repair, tumor cell spread and growth can also be enhanced during liver transection in the donor and later regeneration [18]. In fact, IRI-based inflammation triggers local changes with the development of a favorable microenvironment for tumor cells to invade, migrate and grow.

Moreover, it has been demonstrated in experimental models that the process of liver splitting induces the release of inflammatory molecules as a result of the parenchyma transection, as observed in models of accelerated liver regeneration such as the Associating Liver Partition and Portal vein Ligation for Staged hepatectomy (ALPPS) technique [53,54]. Herein, the authors performed a quantitative polymerase chain reaction (PCR) for interleukin-6 (IL-6) tumor necrosis factor α (TNF-α) (mRNA) in these tissue samples in order to evaluate inflammatory response after ALPPS. Ultimately, these authors demonstrated that liver parenchymal transection dramatically increased the expression of early instigators of regeneration (IL-6 and TNF-α) [55].

Similarly, to the earlier concept of adding a small auxiliary liver graft, known as the concept of APOLT (auxiliary partial orthotopic liver transplantation), a technique that was initially performed based on the concept that the residual native recipient liver would support the overall function of the implanted graft until the graft that is, individually, too small would have grown enough [56].

Another interesting surgical technique to discuss is the living donor (LD) resection and partial liver segment 2–3 transplantation with delayed total hepatectomy (RAPID) procedure initially proposed by Königsrainer et al. [57]. Although experience in HCC candidates is scarce [58], an interesting review article describes 8 LD-RAPID procedures performed in Europe [59]. A clinical bi-institutional prospective study is currently ongoing in Germany to evaluate this technique further for patients with non-resectable colorectal liver metastases. The primary endpoint is a 36-month overall survival (NCT03488953) [59]. This and all other surgical techniques described above would be interesting to be tested in the context of the complex interplay between portal hypertension, liver volume, IRI and cancer recurrence.

Metabolically less favorable livers are often not easily detected just by quantifying the levels of steatosis or considering the age of the donor. Based on this, there is always a risk of developing significant IRI, contributing to the development of SFSS with impaired liver function [60]. The regeneration of the graft might be impaired despite optimal selection and LDLT conditions. This might be the reason why such complications are also observed in donor livers beyond all suggested volume thresholds (GRWR: >0.8–1%) [61,62,63,64,65].

Mitochondrial dysfunction and advanced inflammation in the recipient are the underlying mechanisms of and ongoing inflammation and downstream microvascular tumor spread and recurrence [12,66]. Sometimes, even a low level of ROS released from mitochondrial complex I can be sufficient to trigger this cascade [52]. The direct activation of other cells, including macrophages and endothelial cells, is the next consequence [67].

Provided that the recipient risk is given with the acceptance of the candidate for transplantation and the same is valid for an available living donor, the remaining “wheel to adjust” appears with the preservation of the organ. Early studies have shown that novel dynamic preservation technologies may reduce IRI, reprogram mitochondria and downregulate the recipient’s immune response [68,69,70]. Two main approaches are of interest here: First, the reduction in additional injury, through a replacement of additional cold storage with normothermic machine perfusion (NMP). This technique induces an ex vivo liver IRI and inflammation which is used for viability assessment [71,72,73,74]. The optimal NMP technique is performed from the donor to the recipient center, bridging long cold storage times and the transport [74]. This technically challenging and labor-intensive procedure is tested in various clinical trials with the focus on the utilization of livers with advanced donor risk profiles [71,72,74,75].

The second main technique is the hypothermic oxygenated perfusion (HOPE). Pre-cooled (4–10 °C) artificial fluids are recirculated in a pressure controlled system. Interestingly, the HOPE approach delivering high-perfusate oxygen levels under cold conditions was found to reprogram mitochondria and to prevent later IRI-associated inflammation after liver implantation. Various mechanistic and clinical studies exist together with four randomized controlled trials [62,76,77,78,79,80,81,82]. Based on the underlying mechanism of protection, this perfusion technique reduces posttransplant complications, improves graft survival and reduces re-transplantation rates [76,82]. The concept of improved peripheral sinusoidal flow and microvascular protection through the HOPE concept is also interesting [83,84]. In the context of LDLT, existing experimental studies demonstrated an improved growth and earlier entrance into the cell-cycle for quicker liver regeneration after hypothermic perfusion [85,86,87]. The combination of microvascular protection and better mitochondrial metabolism with more energy (ATP) levels may serve as an additional protective effect inducing better liver function and accelerated regeneration.

Although there are currently no clinical studies on the use of machine perfusion for living donor liver grafts, available experimental studies have triggered the enthusiasm to further assess the HOPE concept. Additional evidence comes from living donor kidney transplantation, where injury occurs during graft procurement which could be reversed with hypothermic perfusion, resulting in better early kidney graft function compared to standard cold storage [87]. Recently, a novel concept of “ischemia-free” organ transplantation (IFOT) has been developed by some authors [88]. The perfusion device is connected to the liver in the donor during procurement and entirely bridges the time spent between the donor and recipient, allowing the ischemic time to be minimal and avoiding cold flush and cold storage. In the first non-randomized clinical trial, the authors from China demonstrated good outcomes with low rates of allograft dysfunction [89]. This concept reduces IRI, inflammation and mitochondrial ROS and DAMPS release with subsequent potential reduction in HCC recurrence. The group from China has assessed the role of IFOT on cancer recurrence and described a reduction compared to standard cold storage [90]. Although such clinical findings support underlying mechanisms, prospective studies are required to gain more information on the role of novel preservation techniques in cancer recurrence and different scenarios with various risk factors.

The present study also highlights the importance of a careful donor selection when considering HCC recipients. Machine perfusion could potentially lead to better graft function, less IRI and downregulate immune response, which is directly linked to cancer recurrence [12,34]. The HOPE-tool could therefore increase the number of available donors for LDLT, considering the high number of steatotic livers in Asian and Arab countries, which mainly rely on LDLT for their candidates, with an ever increasing number of liver cancers.

The present study has several limitations, which should be considered when interpreting the results. The current literature lacks randomized controlled trials, the gold standard study design for comparative studies to provide high-quality evidence. Thus, the best available evidence is based on retrospective observational studies, which are inherently subject to selection bias. These studies consisted of relatively small cohorts and included patients that were treated over a prolonged period of time with more than 20 years. The long period may have introduced additional bias in the context of patient management and modified indications and surgical techniques over time. In addition, donor characteristics (e.g., family history or relation) were not reported in the included studies, which could have caused a bias when interpreting the results. Next, it is also important to highlight that two of the included studies did not report the locoregional liver cancer treatment prior to LDLT. Of note, the identification of prognostic factors appeared to be very challenging due to the lack of standardized parameter reporting. We were therefore not able to develop a formal meta-analysis model.

## 4. Conclusions

The present systematic review has demonstrated that in the setting of LDLT, smaller grafts (i.e., GRWR < 0.8%) could be potentially associated with lower survival rates in HCC recipients, in particular in candidates with tumors outside widely used acceptance criteria for transplantation. A careful donor-recipient selection considering the graft size and liver disease severity with level of portal hypertension for HCC recipients could improve outcomes further.

Future research should target IRI-associated inflammation to understand the relationship between graft size and quality, the ability to regenerate and to handle the relatively high portal flow and subsequent tumor recurrence in the setting of LDLT. Novel dynamic preservation techniques could be of benefit and their role should be explored further.

## 5. Material and Methods

### 5.1. Design and Study Selection

The eligibility criteria, methodology and investigated outcome parameters of the current study were underlined in a review protocol. The latter was registered at the International Prospective Register of Systematic Reviews (registration number: CRD42022384456). The methodology used in the present study respected the standards of Preferred Reporting Items for Systematic Reviews and Meta-Analyses (PRISMA) statement [91].

### 5.2. Eligibility Criteria and Types of Participants

All studies evaluating the outcomes of LDLT comparing the different GRWRs for any adult recipient (aged 18 or older), transplanted for HCC, were considered. Letters, expert opinions, congress abstracts, case series and case reports were excluded. In addition, pediatric transplantation for HCC cases were excluded.

### 5.3. Intervention and Comparison of Interest

LDLT in patients with HCC was considered as the intervention of interest and the different liver GRWRs were evaluated.

### 5.4. Outcomes

The inclusion criteria were studies that reported at least one of the following outcomes: 1-, 3- and 5-year recipient survival and DFS rates (oncological outcomes).

### 5.5. Literature Search Strategy

A comprehensive search strategy was undertaken based on thesaurus headings, search operators and limits in MEDLINE, EMBASE and Web of Science, and was conducted by two independent authors (A.P., A.S.). The last literature search was conducted on December 15th, 2022. A search algorithm that included the terms “liver transplantation” OR “liver transplant” AND “graft recipient weight ratio” OR “graft size” OR “GRWR” AND “hepatocellular carcinoma” OR “liver cancer” OR “HCC” was performed. No limits were set for the year of the publication, but the language was limited to English. Each study identified and included was searched for additional references to identify any further reports or studies of interest that might have been missed.

### 5.6. Selection of Studies

The assessment of the selected studies using title and abstract was conducted by two reviewers, independently (A.P., A.S.). The full texts of relevant articles were collected and evaluated with the eligibility criteria of this study. 

### 5.7. Data Extraction and Management

An electronic data extraction spreadsheet according to Cochrane’s recommendations for intervention reviews was created and was pilot-tested in randomly selected articles and adjusted accordingly. The following information was extracted from each of the included studies by two independent reviewers (A.P., A.S.) to ensure the homogeneity of data collection and to rule out any subjective influence in data collection:study-related data (first author, publication year, country of origin of the corresponding author, journal in which the study was published, study design, procedure performed and sample size of patients in each group);baseline demographic and clinical information of the study populations (donor age, gender, recipient age, recipient MELD, graft type, graft size, graft weight, previous surgical and/or oncological treatment other than transplant, criteria used for transplantation, tumor vascular invasion and GRWR);outcome data.

Disagreements between the two investigators during this process were resolved following iteration, discussion and consultation with a third, independent senior author (K-H.K.). Complete concordance for all variables was achieved.

### 5.8. Assessment of Risk of Bias

The quality of included studies was assessed using the Newcastle–Ottawa Scale (NOS) based on selection (four items), comparability (one item) and outcome (three items) [92]. A nine-star rating system (ranging from 0 to 9) in NOS was used for assessing the quality of observational studies; a study with seven or more stars was regarded as good quality. Conversely, a study with three or more stars but fewer than six was regarded as being of fair quality, whereas two or fewer stars indicated poor quality. Two investigators (A.P., A.S.) reviewed the publications, assessed the quality and extracted the data independently. Disagreements were resolved by discussion and consensus between the two investigators. If no agreement could be reached, a third independent senior author was consulted (K-H.K.). Ultimately, complete concordance was achieved.

### 5.9. Summary Measures, Synthesis, and Statistical Analysis

The potential role of GRWR in the prognosis of patients following LDLT was investigated using 0.8% as the cut-off value based on the reports among available studies. One reviewer (A.P.) independently entered the extracted data into the Review Manager 5.4 software for data synthesis. The entered data were subsequently checked independently by another senior author (A.S.).

## Figures and Tables

**Figure 1 ijms-24-06224-f001:**
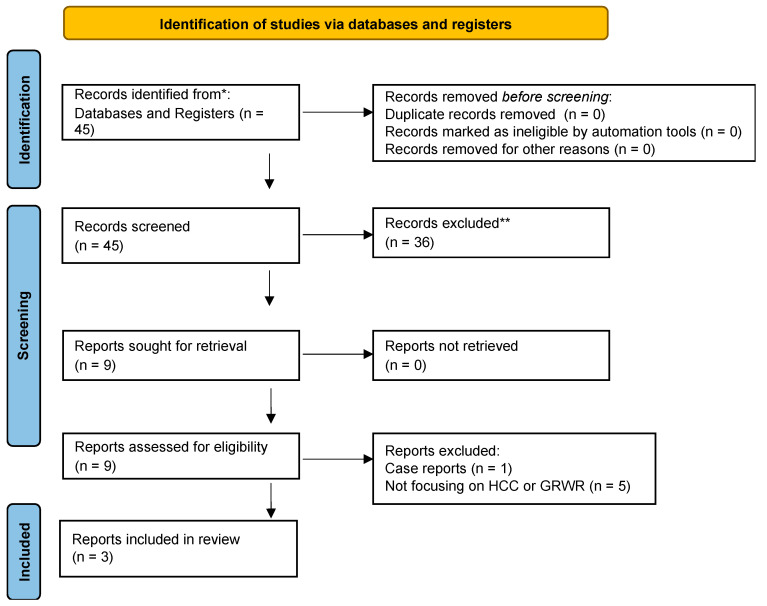
PRISMA flowchart. ***** MEDLINE, EMBASE and Web of Science; ** Sixteen studies were used to give an overview of survival and SFSS rates after LDLT using a cut-off for GRWR at <0.8 % vs. ≥0.8 %. SFSS: small for size syndrome.

**Figure 2 ijms-24-06224-f002:**
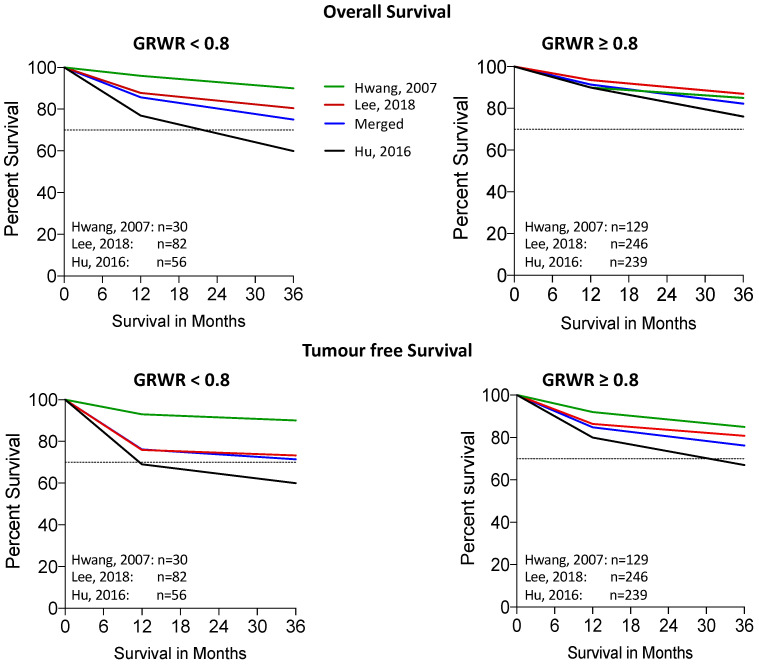
Composite Kaplan–Meier curve plot of overall and HCC-recurrence free survival.

**Figure 3 ijms-24-06224-f003:**
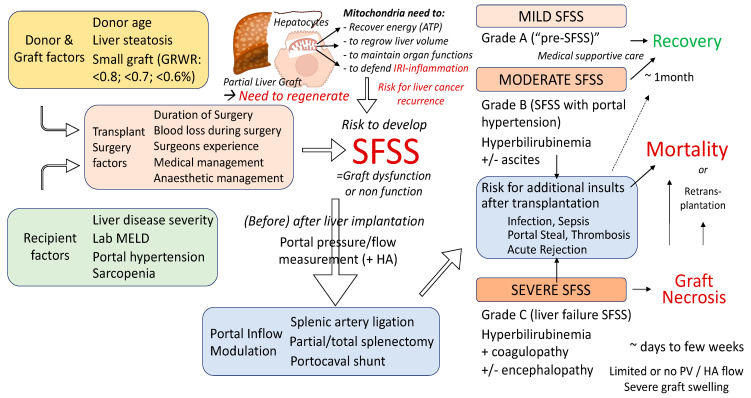
Multiple factors contributing to the development of a small for size syndrome (SFSS) after living donor liver transplantation: Overview on development, risk factors and outcomes of SFSS. In experienced centers, the GRWR can go down to 0.6% and/or GV/SLV ≥ 30% in recipients with low MELD and no significant portal hypertension, related to liver tissue quality. Disturbed blood flows along with elevated levels of IRI lead to an ongoing inflammation, creating a microenvironment that promotes the resettlement and replication of circulating cancer cells, further pronounced by pro-regenerative molecules released to promote liver repair and regeneration.

**Table 1 ijms-24-06224-t001:** Literature overview on outcomes according to accepted liver volume cutoffs for living donor liver transplantation.

Authors, Reference	Year	Country	Study Type	No. of Livers	Lowest Liver Volume				Intermediate Range				Highest Liver Volume			
					GRWR (%)	SFSS Rate (%)	Portal Inflow Modulation	Graft Loss (%)	GRWR (%)	SFSS Rate (%)	Portal Inflow Modulation	Graft Loss (%)	GRWR (%)	SFSS Rate (%)	Portal Inflow Modulation	Graft Loss (%)
Lee et al. [19]	2004	South Korea	Retrospective	79	<0.8	NA	NA	45.4% ^3^	-	-	-		≥0.8	NA	NA	22.1% ^4^
Troisi et al. [20]	2005	Belgium	Retrospective	13	<0.8		HPCS	NA ##	-	-	-		≥0.8			NA ##
Selzner et al. [21]	2009	Canada	Retrospective	337	0.59–0.79	9%	Splenectomy	11 ^1^					≥0.8	2.5%	Splenectomy	9 ^2^
Moon et al. [22]	2010	South Korea	Retrospective	427	<0.8	5.7%	NA	12.2 *	-	-	-		≥0.8	3.6%	NA	9.3 §
Chen et al. [23]	2014	Taiwan	Retrospective	196	<0.8	15.5%	NA	17.8%	-	-	-		≥0.8	5.9%	NA	18.6%
Vasavada et al. [24]	2014	Taiwan	Retrospective	186	<0.8	NA	SAL/ Splenectomy	NA	-	-	-		>0.8	NA	No	NA
Alim et al. [25]	2016	Turkey	Retrospective	649	<0.8	7%	SAL/ Splenectomy	7%	-	-	-		>0.8	11%	None	11
Shoreem et al. [26]	2017	Egypt	Retrospective	174	<0.8	9.8%	Splenectomy	NA	≥0.8–<1	25%	Splenectomy	NA	≥1.0	25%	Splenectomy	NA
Kim et al. [27]	2018	South Korea	Retrospective	160	<0.8	0%	NA	15%	-	-	-		≥0.8	8.3%	NA	8.7%
Goja et al. [28]	2018	India	Retrospective	665	0.55–0.69	10.2%	HPCS/SAL	NA	0.7–0.8	5.4%	SAL	NA	>0.8	1.2%	No	NA
Sethi et al. [29]	2018	India	Retrospective	200	<0.8	12.1%	NA	15.5% ^5^	-	-	-		≥0.8	7.04%	NA	22.8% ^5^
Bell et al. [30]	2018	NA	Meta-analysis	1833	<0.8	10%	NA	NA	-	-	-		≥0.8	5%	NA	NA
Soin et al. [31]	2019	India	Retrospective	1321	0.54–0.69	4.2%	HPCS/SAL	18%	0.7–0.79	2.3%	HPCS/SAL	17%	≥0.8	2.4%	No	14%
Kusakabe et al. [32]	2021	Japan	Retrospective	417	<0.6	20%	Splenectomy	46% ^6^	0.6–≤0.8	20.4%	Splenectomy	21.2% ^6^	>0.8	10.7%	Splenectomy	22.8% ^6^
Jo et al. [33]	2022	South Korea	Retrospective	118	<0.8	27.8%	SAL	13.8%	-	-	-		≥0.8	2%	SA ligation	4%
Wong et al. [34]	2022	Hong Kong	Retrospective	545	≤0.6	12.8%			0.6–≤0.8	13.2%			>0.8	0%		

## Troisi et al. compared GRWR 0.8 with or without porto-caval hemi-transposition, no clear values were therefore available. ^1^—this is extrapolated from “Graft survival at one year 89%”; ^2^—this is extrapolated from “Graft survival at one year 91%”; * this is extrapolated from “Graft survival at one year 87.8%”; § this is extrapolated from “Graft survival at one year 90.7%”; ^3^—this is extrapolated from “Graft survival at one year 54.6%”; ^4^—this is extrapolated from “Graft survival at one year 77.9%”; ^5^—please note this is 90 days mortality, one year not reported.; ^6^—extrapolated from the KM curve and censored; in experienced centers the GRWR can go down to 0.6% and/or GV/SLV ≥ 30% in recipients with low MELD and no significant portal hypertension. GRWR: graft recipient weight ratio; HPCS: hemi-portocaval shunt NA: not available; SAL: splenic artery ligation. SFSS: small-for-size.

**Table 2 ijms-24-06224-t002:** Characteristics of the included studies and patients.

Study	Year	Country	Study Design	Study Period	Number of Patients	Number of Patients with GRWR ≥ 0.8/< 0.8	Donor Mean Age		Recipient Mean Age		Female n (%)		Lab MELD	
							GRWR ≥ 0.8	GRWR < 0.8	GRWR ≥ 0.8	GRWR < 0.8	GRWR ≥ 0.8	GRWR < 0.8	GRWR ≥ 0.8	GRWR < 0.8
Hwang et al. [35]	2007	South Korea	Retrospective	August 1992–December 2004	159	129/30	NA	NA	51.5 ± 6.1	50.3 ± 6.8	25 (19.3)	7 (23.3)	17.7 ± 8.8	16.5 ± 6.4
Hu et al. [36]	2016	China	Retrospective	January 2007–December 2009	295	239/56	26 (23.3–36.5)	28.4 (22.9–41.3)	48.6 (43–54.4)	47.3 (42.7–52.5)	23 (9.6)	1 (1.79)	NA	NA
Lee et al. [37]	2018	South Korea	Retrospective	January 2005–December 2015	328	246/82	33.4 ± 11.7	33.4 ± 12.6	54.7 ± 7.7	53.9 ± 7.6	48 (19.5)	10 (12.2)	11 (9–16)	11 (9–14)

Values are reported as mean ± standard deviation or median (interquartile range) as reported in the original manuscript. NA: not available.

**Table 3 ijms-24-06224-t003:** Graft and tumor characteristics.

Study (First Author [Ref])	Previous Treatment °		Vascular Invasion		Tumor Size (cm)		No. of Nodules		T Stage		Preop. AFP		Graft Type		Graft Weight (g)		Operation Time *		Blood Loss (mL)	
	GRWR ≥ 0.8%	GRWR < 0.8%	GRWR ≥ 0.8%	GRWR < 0.8%	GRWR ≥ 0.8%	GRWR < 0.8%	GRWR ≥ 0.8%	GRWR < 0.8%	GRWR ≥ 0.8%	GRWR < 0.8%	GRWR ≥ 0.8%	GRWR < 0.8%	GRWR ≥ 0.8%	GRWR < 0.8%	GRWR ≥ 0.8%	GRWR < 0.8%	GRWR ≥ 0.8%	GRWR < 0.8%	GRWR ≥ 0.8%	GRWR < 0.8%
Hwang et al. [35]	NA	NA	NA	NA	NA	NA	NA	NA	NA	NA	NA	NA	Right Lobe: 85 (H5678) Left lobe: 15 (H1234) Dual: 29	Right Lobe: 7 (H5678) Left lobe: 20 (H1234) Dual: 3	NA	NA	NA	NA	NA	NA
Hu et al. [36]	108 (45%)	16 (28%)	184 (77%)	44 (78.5%)	3.5 (2.4–5.5)	4 (2.6–5)	1 (1–3)	1 (1–2.5)	NA	NA	107.7 (9.2–1000)	231.6 (9.7–1210)	Right Lobe: 223 (H5678) Left lobe: 4 (H1234) Dual: 1	Right Lobe: 52 (H5678) Left lobe: 3 (H1234) Dual: 0	680 (615–743)	555 (500–607.5)	10.3 (8–12.5)	10.8 (9.5–12.85)	1800 (1000–3000)	2000 (1000–3000)
Lee et al. [37]	NA	NA	104 (42.2%)	41 (50%)	2.2 (1.5–3.5)	2.3 (1.6–3.5)	1 (1–3)	1 (1–2)	I—62 (25.2) II—141 (57.3) III—39 (15.9) IV—4 (1.6)	I—15 (18.3) II—50 (61.0) III—15 (18.3) IV—2 (2.4)	13 (5.1–117.8)	9.6 (3.7–150.4)	Right Lobe: 246 (H5678)	Right Lobe: 82 (H5678)	NA	NA	446.8 ± 121.2	447.8 ± 105.1	1550 (800–3500)	1500 (800–3000)

Values are reported as mean ± standard deviation or median with interquartile range as reported in the original manuscript. ° Includes previous partial hepatectomy (9 cases for GRWR ≥ 0.8; 1 case for GRWR < 0.8), RFA (14 cases for GRWR ≥ 0.8; 2 cases for GRWR < 0.8), trans-arterial chemoembolization (64 cases for GRWR ≥ 0.8; 2 cases for GRWR < 0.8), percutaneous ethanol injection (1 case for GRWR ≥ 0.8; 11 cases for GRWR < 0.8), combined treatments (17 cases for GRWR ≥ 0.8; 2 cases for GRWR < 0.8); * Operation time is reported in hours for Hu et al. and in minutes for Lee et al. AFP: alpha-fetoprotein. NA: not available.

**Table 4 ijms-24-06224-t004:** Summary of risk of bias assessment using NOS system.

	Number of Stars			
Study (Year)	Selection *	Comparability #	Outcome °	Overall
Hwang 2007	3	2	2	7/9
Hu 2016	3	1	2	6/9
Lee 2018	3	2	2	7/9

* Maximum 4 stars; # Maximum 2 stars; ° Maximum 3 stars.

## Data Availability

The data presented in this study are openly available.

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
