# Peer review of "Association between Hepatocellular Carcinoma Recurrence and Graft Size in Living Donor Liver Transplantation: A Systematic Review"

_ijms, 2023, doi:10.3390/ijms24076224_

Round 1

Reviewer 1 Report

Dear Authors,

Certainly, I can review the article "Association Between Hepatocellular Carcinoma Recurrence and Graft Size in Living Donor Liver Transplantation: A Systematic Review" in the context of journal.

The article is well-written and provides a comprehensive analysis of the available literature on the relationship between graft size and hepatocellular carcinoma (HCC) recurrence in living donor liver transplantation. The authors conducted a thorough search of multiple electronic databases and used a clear set of inclusion and exclusion criteria to identify relevant studies for inclusion in their systematic review.

Kind regars,

Author Response

Certainly, I can review the article "Association Between Hepatocellular Carcinoma Recurrence and Graft Size in Living Donor Liver Transplantation: A Systematic Review" in the context of journal.

The article is well-written and provides a comprehensive analysis of the available literature on the relationship between graft size and hepatocellular carcinoma (HCC) recurrence in living donor liver transplantation. The authors conducted a thorough search of multiple electronic databases and used a clear set of inclusion and exclusion criteria to identify relevant studies for inclusion in their systematic review.

Kind regars,

Our Reply:

We thank the reviewer for the kind evaluation of our manuscript.

Reviewer 2 Report

Alessandro et al reported a potentially interesting finding on the GRWR and the recurrence of HCC in adult LDLT patients. The topic and the findings are of great clinical significance, yet there are several flaws that required further clarification.

1. The study should report the base level of the HCC patients with more details. For example, the TNM staging, AFP level, tumor size, metastasis, chemotherapy history or another treatment history, etc. 

2. The surgical parameters of all patients should be shown in the paper. For example, the time of the surgery, hemorrhage volume, etc.

3. Donor characteristics should be more detailed. The authors only focused on the gender and the weight of the graft, yet in my opinion, the baseline features of the donor could significantly affect the prognosis of the recipient. For example, the recipient’s and donor’s family history, if the donors would develop malignancies after LDLT, etc. 

4. It seems that in figure2, there is quite some difference in the survival and recurrence of HCC among different centers. And the difference between Huang et al and other groups is even bigger than the difference between GRWR. Is this caused by the surgery or post-surgery treatment (such as the dosage of different immunosuppressive drugs or steroids)? How would the authors interpret these differences?

Author Response

Alessandro et al reported a potentially interesting finding on the GRWR and the recurrence of HCC in adult LDLT patients. The topic and the findings are of great clinical significance, yet there are several flaws that required further clarification.

Our Reply:

We thank the reviewer for the kind evaluation of our manuscript and the very valuable suggestions, which we address below.

  1. The study should report the base level of the HCC patients with more details. For example, the TNM staging, AFP level, tumor size, metastasis, chemotherapy history or another treatment history, etc. 

Our Reply:

We thank the reviewer and have modified the Tables accordingly. The level of preoperative AFP and other requested parameters (i.e., including TNM stage, tumor size, metastasis and pre transplant treatments ) were added to Table 3 and we describe such details in the text, although not all studies provide the complete set of parameters. Following the reviewer’s comments, we have added further details regarding previous treatment in the capture of Table 3.   

  1. The surgical parameters of all patients should be shown in the paper. For example, the time of the surgery, hemorrhage volume, etc.

Our Reply:

We thank the reviewer for this comment. Available data in the included studies have been added to Table 3.

  1. Donor characteristics should be more detailed. The authors only focused on the gender and the weight of the graft, yet in my opinion, the baseline features of the donor could significantly affect the prognosis of the recipient. For example, the recipient’s and donor’s family history, if the donors would develop malignancies after LDLT, etc. 

Our Reply:

We thank the reviewer for raising this important point. Unfortunately, these data are not available in the included studies used for the systematic review. Therefore, we were not able to include such donor details, although we believe that the reviewer has highlighted an important point. We have amended the limitation section in the revised discussion.

  1. It seems that in figure2, there is quite some difference in the survival and recurrence of HCC among different centers. And the difference between Huang et al and other groups is even bigger than the difference between GRWR. Is this caused by the surgery or post-surgery treatment (such as the dosage of different immunosuppressive drugs or steroids)? How would the authors interpret these differences?

Our Reply:

We thank for this important comment and agree. The study from Hwang et al was conducted in Asan Medical Center which is well known for their vast experience in LDLT. This study was published in 2007 with older cases, done between August 1992 and Dec 2004, a period where LDLT and the experience was still growing. It is therefore possible that the team was more conservative in their donor and recipient selection, although we may here only speculate because the authors did not report the donor age. The GRWR of <0.8 in studies other than Hwang et al may well be <0.7, compared to Hwang et al, where the GRWR could be closer to 0.8 (i.e., between >0.7 and <0.8), leading to better outcomes due to more liver volume. We have added a few details in the discussion, please see the third paragraph of the revised discussion. Unfortunately, details on the posttransplant medical management and immunosuppression were not presented either and we were not able include such parameters. It is possible that multiple factors have played a role.

Reviewer 3 Report

We would like to congratulate the authors for this interesting review on an important subject. Please find our comments and suggestions below

Methods and Results:

-          For more clarity, please use the New World Terminology to describe the grafts used in the study

-          Lee et al state in their study that they included only right lobe grafts on their study. Please correct in the table and the text.

-          Please also specify the strategy of the different authors regarding the middle hepatic vein

-          It would be interesting to add data from studies on SFSS in partial grafts from deceased donors as comparator if available given that these grafts are more prone to ischemia-reperfusion injury.

Discussion:

-          The main finding of the study namely that Milan out (not Milan in) and occurrence of SFSS have impaired oncological outcomes is not discussed. What is a potential hypothesis for this observation?

-          Could the authors also discuss the important differences in survival observed between the study of Hwang et al and the 2 others especially for grafts with a GRWR < 0.8. Is the fact that Hwand et al used predominantly left lobe grafts a possible factor? In this context what ay be the impact of left or right lobe grafts on SFSS and recurrence?

-          Overall, it seems important to clearly distinguish several pathophysiological mechanisms described in this paper.  SFSS is a result of a too high portal inflow and too little functional liver remnant leading to impaired regeneration and is and not necessarily related to IRI. Liver grafts presenting SFSS may be more susceptible to IRI but it seems that those are 2 different mechanistic entities. In this context, the authors should deemphasize their discussion on known mechanisms of IRI of whole grafts mediated by ROS release upon reperfusion and provide a more focused discussion on the specificities of partial grafts in regard to SFSS, inflow and outflow and regeneration. As the author point out in their discussion, grafts with a GRWR > 0.8% may also develop severe IRI but do not present SFSS and subsequently higher recurrence rate.

-          Could the authors discuss how the presented data must be interpreted in line with the paradigm to use smaller grafts in order to shift the risk of the LD procedure from the donor to the recipients. Is this strategy still valid or should be move back to using right partial grafts?

-          It is unclear in the manuscript how machine perfusion and especially normothermic machine perfusion may benefit small LDLT graft and prevent SFSS. Please reduce the paragraph on machine perfusion as the presented data are very preliminary.

-          There are other important possible “wheels to adjust” in the use of small grafts in LDLT as recently reported by Fujiki et al such as inflow modulation and outflow optimization to reduce occurrence of SFSS. These aspects should also be discussed

-          Could the use of RAPID procedure prevent HCC recurrence?Please discuss

Author Response

We would like to congratulate the authors for this interesting review on an important subject. Please find our comments and suggestions below

Methods and Results:

-          For more clarity, please use the New World Terminology to describe the grafts used in the study

Our reply:

We thank the reviewer and have added the new terminology suggested by Nagino et al in 2021 with references number 37. Right and left lobe hepatectomies for LDLT would classify as “RLG (H5678)” and “LLG (H1234-MHV)”, respectively. The challenge with the further specification is that the included studies do not further specify if in a few cases liver lobes were resected formally other than the standard left or right hepatectomy. We have added this classification and the reference in the revised manuscript.

-          Lee et al state in their study that they included only right lobe grafts on their study. Please correct in the table and the text.

Our Reply:

We thank the reviewer and have adapted the Table and text accordingly.  Please see the revised Table 3.

-          Please also specify the strategy of the different authors regarding the middle hepatic vein

Our Reply:

We thank the reviewer for these important comments. Interestingly two studies did not discuss the strategy for the management of the middle hepatic vein. Lee et al have reported that any middle vein branch of >5 mm in diameter was saved for reconstruction. If an inferior right hepatic vein branch of >5 mm was present following right hepatic vein anastomosis, this branch was anastomosed to the inferior vena cava as well. We have amended the result section accordingly.

-          It would be interesting to add data from studies on SFSS in partial grafts from deceased donors as comparator if available given that these grafts are more prone to ischemia-reperfusion injury.

Our Reply:

We thank the reviewer for this important point. We agree that partial grafts from deceased donors (i.e., split grafts) are at higher risk for reperfusion injury, which could be an interesting topic in context of initial dysfunction (i.e., PNF or severe EAD) and also HCC recurrence. Based on strict selection criteria for the acceptance of deceased donors for split procedures in all countries, the risk for a SFSS (which is not really used for this type of grafts) is not well assessed. In contrast, the use terminology is more on the side of deceased grafts than LDLT. The highest risk is to suffer vascular complications (i.e., arterial issues), bleedings and bile leaks from the cut surface. The context to SFSS/PNF and also HCC recurrence is not well explored, certainly only a few cases, particularly when looking into left lobe transplants from deceased donors. A few reports exist

in the literature (PMID: 10973381, PMID: 11303140), which are however quite old (year 2000) and do not compare whole deceased with split grafts or even living donors. Although we agree on this interesting point, we were not able to include split grafts into our systematic review because of a lack of studies in this context. However, we have added a few points raised into the discussion. Please see the revised manuscript and thank you again for this interesting comment.

Discussion:

-          The main finding of the study namely that Milan out (not Milan in) and occurrence of SFSS have impaired oncological outcomes is not discussed. What is a potential hypothesis for this observation?

Our reply:

We thank the reviewer for the suggestion and have added more details to our discussion. The underlying mechanism in accordance with the available literature is an advanced recipient cancer status with more aggressive tumour biology combined with smaller grafts at risk or with features of a SFSS, triggering a more accelerated liver growth, which in turn triggers more cell replication seen also in tumour cells, naturally leading to higher recurrence rates. In the literature there is a large body showing a higher HCC recurrence rate with livers transplanted from deceased donors with higher risk, including DCD grafts or longer cold ischemia time. At the same time an elevated HCC risk on the recipient side does also trigger more recurrence. The combination, though not well assessed with clinical data, would be expected to even trigger more recurrence.

-          Could the authors also discuss the important differences in survival observed between the study of Hwang et al and the 2 others especially for grafts with a GRWR < 0.8. Is the fact that Hwand et al used predominantly left lobe grafts a possible factor? In this context what may be the impact of left or right lobe grafts on SFSS and recurrence?

Our reply:

We thank the reviewer for this point, also raised by another reviewer. The study from Hwang et al was conducted in Asan Medical Center, which is well known for their vast experience in LDLT. This study was published in 2007 with older cases, done between August 1992 and Dec 2004, a period where LDLT and the experience was still growing. It is therefore possible that the team was more conservative in their donor and recipient selection, although we may here only speculate because the authors did not report the donor age. The GRWR of <0.8 in studies other than Hwang et al may well be <0.7, compared to Hwang et al, where the GRWR could be closer to 0.8 (i.e., between >0.7 and <0.8), leading to better outcomes due to more liver volume. We have added a few details in the discussion, please see the third paragraph of the revised discussion. Unfortunately, details on the posttransplant medical management and immunosuppression were not presented either and we were not able include such parameters. It is possible that multiple factors have played a role. Unfortunately, the included studies do not specify if more recurrence was seen with left or right lobe donors. We have amended the discussion by interpreting the better results from Hwang et al. and are highlighted in the discussion.

-          Overall, it seems important to clearly distinguish several pathophysiological mechanisms described in this paper.  SFSS is a result of a too high portal inflow and too little functional liver remnant leading to impaired regeneration and is and not necessarily related to IRI. Liver grafts presenting SFSS may be more susceptible to IRI but it seems that those are 2 different mechanistic entities. In this context, the authors should deemphasize their discussion on known mechanisms of IRI of whole grafts mediated by ROS release upon reperfusion and provide a more focused discussion on the specificities of partial grafts in regard to SFSS, inflow and outflow and regeneration. As the author point out in their discussion, grafts with a GRWR > 0.8% may also develop severe IRI but do not present SFSS and subsequently higher recurrence rate.

Our reply:

We thank the reviewer for this great suggestion. We agree, that SFSS is not necessarily related with IRI but depends on the liver volume related with portal hypertension and liver in- and outflow. We have added further details and differentiate between the potential effect of such features and IRI and the underlying mechanisms. Please see the revised discussion. We have also specified the role of IRI comparing deceased donor and living donor grafts, providing more information on mechanisms of SFSS, in/outflow and regeneration and how such features communicate on the level of mitochondria. 

-          Could the authors discuss how the presented data must be interpreted in line with the paradigm to use smaller grafts in order to shift the risk of the LD procedure from the donor to the recipients. Is this strategy still valid or should be move back to using right partial grafts?

Our reply:

We thank the reviewer for this comment. A careful donor selection remains important for the safety of the donor and will always be highly relevant. Although the use of right lobe grafts could translate into a higher risk for the donor, the GRWR thresholds should be considered, which might be one reason why centres in Korea have conservative guidelines regarding the GRWR cut-off to sustain their excellent outcomes. Other potential techniques (e.g. dual grafts) are considered to avoid otherwise too risky situations thereby reducing the rate of SFSS and avoid too smaller liver volumes. The required volume should however be selected as small as possible to protect donors. Centers will practice further according to their experience, the required donor volume also depends on the disease severity of the recipient in particular on the level of portal hypertension.

-          It is unclear in the manuscript how machine perfusion and especially normothermic machine perfusion may benefit small LDLT graft and prevent SFSS. Please reduce the paragraph on machine perfusion as the presented data are very preliminary.

Our reply:

We thank the reviewer for this comment. We agree that normothermic machine perfusion may not been beneficial in this context. Indeed, in the literature there is enough mechanistical evidence for the mitochondrial protection by hypothermic oxygenated perfusion (HOPE) demonstrating better liver regeneration, quicker cell cycle entrance by hepatocytes with HOPE. Based on this we have included such options in the discussion. Another advantage is that HOPE is only performed through the portal vein without touching the hepatic artery. The number of studies showing the protection of partial deceased graft recipients is also increasing. A positive side effect of HOPE is the reduced reperfusion injury after transplantation and the better immediate liver function, potentially contributing to the avoidance of SFSS or the reduction of the severity. However this needs to be studies systematically and a clinical trial of the impact of HOPE on SFSS is currently started in a very experienced LDLT centre in India.  

-          There are other important possible “wheels to adjust” in the use of small grafts in LDLT as recently reported by Fujiki et al such as inflow modulation and outflow optimization to reduce occurrence of SFSS. These aspects should also be discussed.

Our reply:

We thank the reviewer and have added more details in this regard into the revised discussion also in context of the mechanistical figure. The manuscript by Fujiki et al is a great paper particularly as it is written by a group from a primarily not Asian country.

-          Could the use of RAPID procedure prevent HCC recurrence? Please discuss

Our reply:

We thank the reviewer for this question. The RAPID procedure was initially developed to offer transplantation for unresectable colorectal metastasis in situations where no deceased donor is available or together with other challenges in countries with long waiting list times. Although the RAPID concept could be of interest and is increasingly explored, expected challenges are quite comparable to small living donor grafts. With the rapid procedure a partial deceased donor graft or segmental living donor graft (i.e., segment 2+3) is implanted together with a reduction of the recipient’s own liver, while the right lobe remains in situ to support the overall liver function. In patients with advanced portal hypertension the flow division between right recipient lobe and newly transplanted left lobe segments is a challenge and recipients are highly selected, also in context of the tumour biology. To our best knowledge, 8 LD- RAPID were performed so far (5 in Germany, 2 in Italy, and 1 in Belgium). Of note, three of eight patients are alive and tumor-free after 6–18 months (PMID: 33740114). A prospective clinical trial is currently done in Germany (Tübingen and Jena) to evaluate the feasibility, safety, and efficacy of LD-RAPID for non-resectable CRLM (NCT03488953). Although this is an interesting approach the data on HCC cases are scarce (PMID: 33197130) and more publications are awaited. With a small additional graft and diverted portal flow the growth trigger would be expected to be lower compared to the direct implantation of a partial deceased or living donor graft. The also relevant level of IRI comparing partial living and deceased donor grafts is less well assessed, and data on HCC recurrence are even more scarce. We have added the RAPID procedure and related discussion in the revised manuscript and made reference to current clinical studies.

Round 2

Reviewer 2 Report

My concerns are well addressed. No further questions will be asked.

Author Response

Reviewer #2

My concerns are well addressed. No further questions will be asked.

Our Reply:

We thank the reviewer for the kind evaluation of our revised manuscript.

Reviewer 3 Report

We thank the authors for providing an excellent revised version of there manuscript.

I have one last very minor comment : please use the new world terminology in the entire manuscript in order to stansardize the terminology of partial grafts.

Author Response

Reviewer #3

We thank the authors for providing an excellent revised version of there manuscript.

I have one last very minor comment : please use the new world terminology in the entire manuscript in order to stansardize the terminology of partial grafts.

Our reply:

We thank the reviewer for the thorough assessment of our revised manuscript. We have added the new terminology in the tables and in the discussion. These amendments are highlighted in the new revised version.